# Epimesatines P–S: Four Undescribed Flavonoids from *Epimedium sagittatum* Maxim. and Their Cytotoxicity Activities

**DOI:** 10.3390/molecules29194711

**Published:** 2024-10-04

**Authors:** Shuang-Shuang Xie, Xiang Yu, Jing-Ke Zhang, Zhi-You Hao, Xiao-Ke Zheng, Wei-Sheng Feng

**Affiliations:** 1School of Pharmacy, Henan University of Chinese Medicine, Zhengzhou 450046, China; shuangxie@hactcm.edu.cn (S.-S.X.); yuxiang_0818@163.com (X.Y.); 18137802812@163.com (J.-K.Z.); hzy@hactcm.edu.cn (Z.-Y.H.); zhengxk.2006@163.com (X.-K.Z.); 2The Engineering and Technology Center for Chinese Medicine Development of Henan Province, Zhengzhou 450046, China; 3Co-Construction Collaborative Innovation Center for Chinese Medicine and Respiratory Diseases by Henan & Education Ministry of China, Zhengzhou 450046, China

**Keywords:** *Epimedium sagittatum* Maxim., flavonoids, electronic circular dichroism (ECD), MCF-7, sphingosine kinase 1 (Sphk1)

## Abstract

In this study, four previously undescribed flavonoids, named epimesatines P (**1**), Q (**2**), R (**3**), and S (**4**), were isolated from the aerial parts of *Epimedium sagittatum* Maxim. Their structures and absolute configurations were confirmed via spectroscopic analyses, quantum chemical electronic circular dichroism (ECD) calculations, Mo_2_(OAc)_4_–induced ECD, and Rh_2_(OCOCF_3_)_4_–induced ECD experiments. Epimesatines Q and R were characterized by the presence of furan rings. A cytotoxicity assay demonstrated that epimesatines P–S exhibited significant inhibitory effects on the viability of MCF-7 human breast cancer cells, with IC_50_ values ranging from 1.27 to 50.3 μM. Notably, epimesatines Q and R exhibited superior efficacy against MCF-7 cells compared to epimesatines P and S, suggesting that the presence of furan rings may enhance their activity against MCF-7 cells. Specifically, epimesatine Q displayed a more potent inhibitory effect at 1.27 μM compared to a positive control, docetaxel, which had an IC_50_ of 2.13 μM, highlighting its potential as a therapeutic agent for breast cancer. Importantly, none of the tested compounds exhibited obvious toxicity toward MCF-10A human breast epithelial cells. Furthermore, compounds **1**, **3**, and **4** were found to significantly inhibit the expression of sphingosine kinase 1 (Sphk1) in MCF-7 cells.

## 1. Introduction

Epimedii Folium, commonly known as Yin Yang Huo in Chinese, is a plant belonging to the Berberidaceae genus that has been traditionally used for addressing sexual dysfunction, cardiovascular diseases, osteoporosis, chronic nephritis, and asthma [1,2,3,4]. Exploration of the phytochemical composition of Epimedium species dates back to 1935 by Akai et al. [5], and flavonoids, phenols, lignans, ionones, and alkaloids have been identified from this species [6,7,8,9,10]. The 2020 edition of Chinese Pharmacopoeia [11] officially recognizes *Epimedium sagittatum* Maxim., *E. koreanum* Nakai., *E. brevicornum* Maxim., and *E. pubescens* Maxim. as the primary sources of Herba Epimedii, with *E. sagittatum* being the most commonly utilized due to its widespread availability [12]. *E. sagittatum* is distributed across various regions in China, including Anhui, Guangxi, Fujian, Guizhou, Hubei, and Hunan, primarily growing in forests, water ditches, shrubs, or rock crevices at altitudes ranging from 200 to 1750 m [1,7]. The plant has been extensively utilized for the treatment of sexual dysfunction, osteoporosis, cardiovascular diseases, asthma, and chronic nephritis [1,13]. Previous studies have identified *E. sagittatum* as a source of diverse chemical constituents, including flavonoids, lignanoids, ionones, phenol glycosides, phenylethanoid glycosides, and sesquiterpenoids [13]. Notably, flavonoids, which are promising constituents for cancer therapeutics [14], have been recognized as the primary bioactive components of this plant [1,13]. *E. sagittatum* has demonstrated significant anti-tumor potential against various cancer cells, including lung cancer, breast cancer, colorectal cancer, hepatocellular carcinoma, esophageal cancer, and others [15,16,17]. Notably, icariin, a flavonoid derived from *E. sagittatum,* has been developed as a commercial anti-hepatocellular carcinoma drug [18]. Consequently, *E. sagittatum* represents a promising plant source for the development of anti-tumor drugs.

Cancer remains a significant global health concern, and chemotherapy is a crucial treatment approach. However, the side effects and drug resistance associated with chemical drugs underscore the necessity for exploring new agents with anti-tumor properties [19,20]. Medicinal plants are considered promising resources for natural products with anti-tumor properties due to their extensive history of medicinal use and abundant availability [21,22,23,24].

Given the potential for deriving anti-tumor precursors from *E. sagittatum*, this study aims to look for structurally intriguing flavonoids with anticancer effects sourced from this plant. Consequently, four previously undescribed flavonoids, named epimesatines P (**1**), Q (**2**), R (**3**), and S (**4**), were isolated and identified. Notably, epimesatines Q and R are two flavonoids that infrequently incorporate furan rings in their structures. The viability of the isolated compounds on MCF-7 human breast cancer cells, A549 human lung carcinoma cells, and SMMC-7721 human hepatoma cells was assessed using an MTT assay. Furthermore, the influence of **1**–**4** on the expression levels of sphingosine kinase 1 (Sphk1), which is an enzyme implicated in cancer progression [25], was investigated in MCF-7 cells. Herein, the isolation, structural elucidation, and biological activity evaluation of these metabolites are described (see Figure 1).

## 2. Results and Discussion

### 2.1. Structure Elucidation

Epimesatine P (**1**) was isolated as a yellow powder. The molecular formula of **1** was established as C_25_H_26_O_8_ through high-resolution electrospray ionization mass spectrometric (HRESIMS) (*m*/*z* 477.1509 [M + Na]^+^, calcd. for 477.1520) and ^13^C nuclear magnetic resonance (NMR) data, corresponding to thirteen degrees of unsaturation. Analysis of the ^1^H NMR data of **1** (Table 1, Appendix A) revealed three methyl singlets (*δ*_H_ 1.83, 1.36, and 1.19), three methylene groups (*δ*_H_ 4.42 and 3.33, each 2H; 4.90 and 4.75, each 1H), and five olefinic protons (*δ*_H_ 7.75, 7.74, 6.60, 6.54, and 6.25, each 1H). The ^13^C NMR (Table 1, Appendix A) and DEPT-135 data of **1** indicated 25 carbon signals, including three methyls (*δ*_C_ 21.5, 19.6, and 18.1), three methylenes (*δ*_C_ 110.6, 37.2, and 31.1), seven methines (including two oxygenated, *δ*_C_ 87.4 and 75.2), and twelve quaternary carbons (including one ketone carbonyl, *δ*_C_ 182.9, and one oxygenated, *δ*_C_ 83.0). Based on the above data, compound **1** was identified as a prenylated flavone derivative [26,27,28]. The elucidation of a 2-hydroxy-3-methylbut-3-enyl moiety was deduced through ^1^H−^1^H correlation spectroscopy (COSY) correlation of H_2_-1′′′/H-2′′′ and heteronuclear multiple bond correlation (HMBC) correlations from Me-5′′′ to C-2′′′, C-3′′′, and C-4′′′ (Figure 2). Furthermore, key HMBC correlations from H_2_-1′′′ to C-4′, C-5′, and C-6′ verified the substituent at C-5′. In addition, HMBC correlations from Me-4″/Me-5″ to C-2″ and C-3″, along with the spin coupling system of H_2_-1″/H-2″, established a 2,3-dioxygenated prenyl unit. Considering the molecular formula, the chemical shifts of C-2″ (*δ*_C_ 87.4) and C-3″ (*δ*_C_ 83.0), and the key HMBC correlations from H_2_-1″ to C-2′, C-3′, and C-4′, as well as from H_2_-2″ to C-4′, a pyran ring was formed across C-3′ and C-4′, with C-3″ being oxygenated by -OOH. Consequently, the planar structure of compound **1** was established as shown.

The absolute configuration of **1** was ascertained through electronic circular dichroism (ECD) calculations using time-dependent density functional theory methodology at the B3LYP/6-31G(d) level [29,30]. The good consistency of the calculated and experimental ECD curves of **1** (Figure 3) indisputably established its absolute configuration as 2″S,2′′′S.

Epimesatine Q (**2**) was obtained as a yellow powder, and its molecular formula was C_29_H_34_O_9_ according to the HRESIMS data (*m*/*z* 549.2083 [M + Na]^+^, calcd. for 549.2095), indicating thirteen degrees of unsaturation. Through a comprehensive analysis of its ^1^H and ^13^C NMR data (Table 1), compound **2** was also classified as a flavonoid. Analysis of HMBC correlations (Figure 2) from Me-4′′′/Me-5′′′ to C-2′′′ (*δ*_C_ 78.3) and C-3′′′ (*δ*_C_ 72.9), along with the ^1^H-^1^H COSY correlation (Figure 2) of H_2_-1′′′/H-2′′′, led to the deduction of a 2,3-dihydroxy-3-methylbutyl unit. Moreover, the HMBC correlations from H_2_-1′′′ to C-4′, C-5′, and C-6′ confirmed this substitute at C-5′. The HMBC correlations from Me-4″/Me-5″ to C-2″ (*δ*_C_ 71.6) and C-3″ (*δ*_C_ 78.4), and from H_2_-1″ to C-2′, C-3′, and C-4′, along with the ^1^H-^1^H COSY correlation of H_2_-1″/H-2″, indicated the existence of a 2-hydroxy-3,3-dimethyldihydropyran ring bridging C-3′ and C-4′ [31]. Additionally, a furan ring was elucidated using the spin coupling system of H_2_-1′′′′/H_2_-2′′′′/H_2_-3′′′′/H_2_-4′′′′, and the location of the furan ring was confirmed based on HMBC correlation from H_2_-1′′′′ to C-2″. Consequently, the planar structure of **2** was elucidated.

Due to the existence of a vicinal diol moiety in the side chains of **2**, Mo_2_(OAc)_4_–induced ECD experiments were performed to determine the absolute configuration of C-2′′′. Accordingly, the configuration of C-2′′′ was established as *S* based on the positive Cotton effect at 312 nm in the Mo_2_(OAc)_4_–induced ECD spectrum (Figure 4), following the helicity rule [32,33]. Additionally, the absolute configuration of **2** was determined as 2″R,2′′′S,1′′′′R through the comparison of experimental and calculated ECD curves (Figure 3).

The molecular formula of epimesatine R (**3**) was identified as C_29_H_34_O_10_ based on the HRESIMS data (*m*/*z* 565.2036 [M + Na]^+^, calcd. for C_29_H_34_O_10_Na, 565.2044), indicting thirteen degrees of unsaturation. Detailed analysis of the NMR data revealed structural similarities between compounds **3** and **2**, except that a hydroxy at C-3′′′ in **2** was replaced by a hydroperoxy in **3**. This deduction was supported by the notable upfield chemical shift of C-3′′′ (*δ*_C_ 86.0) and the molecular mass of **3**, which was 16 Da higher than **2**, indicating the existence of an additional oxygen atom in **3**. Thus, the structure of **3** was established as depicted. The absolute configuration of C-2″ in **3** was ascertained through Rh_2_(OCOCF_3_)_4_–induced ECD experiments, revealing an *S* configuration based on the negative Cotton effect at 351 nm (Figure 4). Finally, the absolute configuration of **3** was determined as 2″R,2′′′S,1′′′′R based on the alignment of experimental and calculated ECD curves (Figure 3).

Epimesatine S (**4**) was determined to have a molecular formula of C_26_H_28_O_5_ based on the HRESIMS ion at *m*/*z* 421.2006 [M + H]^+^ (calcd. for C_26_H_29_O_5_, 421.2010). Detailed analysis of the 1D NMR data suggested that compound **4** closely resembled 5,7,4′-trihydroxy-8,3′-diprenylflavone [5,34], except for the presence of an additional methoxy signal in **4**, which was supported by HMBC correlation from OCH_3_ (*δ*_H_ 3.97) to C-4′. Thus, the structure of **4** was established as shown in Figure 1.

### 2.2. In Vitro Cytotoxicity

Since the primary objective of this study was to discover compounds with anti-tumor effects, the cytotoxic activity of compounds **1**–**4** was evaluated against MCF-7, A549, and SMMC-7721 cancer cells through the MTT method [35,36]. As illustrated in Figure 5a, compounds **1**–**4** exhibited a statistically significant reduction (*p* < 0.01) in the viability of MCF-7 cells at concentration of 10 μM when compared to the control (CON) group. This observation indicated that these compounds may serve as the active constituents of *E. sagittatum* in exerting anti-breast cancer activity. Subsequently, the evaluation of cytotoxic effects on MCF-10A human breast epithelial cells revealed no significant difference when compared to the CON group (Figure 5b), indicating the safety of these compounds for normal human breast cells. Therefore, these compounds hold promise for the development of anti-breast cancer drugs. The activity of compounds **1**–**4** against A549 and SMMC-7721 cancer cells was evaluated using the same method. However, none of these compounds exhibited significant cytotoxicity towards either of the cancer cell lines (Figure 5c,d).

Subsequently, the expression levels of Sphk1 in MCF-7 cells treated with compounds **1**–**4** were evaluated using immunofluorescence [37]. The findings, as illustrated in Figure 6, revealed that compounds **1**, **3**, and **4** effectively inhibit (*p* < 0.01) the expression of Sphk1 in MCF-7 cells compared to the CON group, suggesting that Sphk1 may represent a viable therapeutic target for breast cancer in relation to these compounds. However, the fluorescence intensity observed with compound **2** did not exhibit a significant difference when compared to the CON group, suggesting that compound **2** may exert its effects on MCF-7 breast cells via other pathways.

A real-time cellular analysis (RTCA) assay [38] indicated that the IC_50_ values of compounds **1**–**4** on MCF-7 cells ranged from 1.27 to 50.3 μM, with compounds **2** and **3** demonstrating superior effects compared to compounds **1** and **4** (Table 2). This observation suggests that the presence of furan rings may enhance the activity of these compounds on MCF-7 cells. Specifically, compound **2** (1.27 μM) displayed a stronger inhibitory effect on MCF-7 cell viability than the positive control, docetaxel (2.13 μM), further supporting its potential as a therapeutic agent against breast cancer.

## 3. Materials and Methods

### 3.1. General Experimental Procedures

Optical rotations and CD spectra were acquired by a Rudolph AP-IV polarimeter (Rudolph, Hackettstown, NJ, USA) and a Chirascan qCD spectrometer (Applied Photophysics Ltd., Surrey, UK), respectively. UV spectra were collected on an Evolution 300 instrument (Thermo, Waltham, MA, USA). IR spectra were measured on a Nicolet IS 10 spectrophotometer (Thermo, Waltham, MA, USA). HRESIMS data were obtained in positive-ion mode on a Bruker Maxis HD mass spectrometer (Bruker, Berlin, Germany). NMR spectra were measured on a Bruker Advance III 500 spectrometer (Bruker, Berlin, Germany), and chemical shifts were referenced to residual solvent signals. Silica gel (100–200 and 200–300 mesh, Marine Chemical Industry, Qingdao, China) and ODS (50 μM, YMC Group, Kyoto, Japan) were used for column chromatography (CC). Semipreparative high-performance liquid chromatography (HPLC) separations were performed on a Shimadzu LC-40 HPLC system, equipped with a DAD detector, using a reversed-phase (RP) C18 ODS column (10ID × 250 mm, Cosmosil 5C18-MS-II Packed column, Nacalai Tesque, Tokyo, Japan). The analytical solvents (EtOH, MeOH, CH_2_Cl_2_, petroleum ether, and EtOAc) and chromatographic (high-purity MeOH and MeCN) solvents employed in this study were purchased from Tianjin Fuyu Fine Chemical Co., Ltd. (Tianjin, China) and Anhui Tiandi High Purity Solvent Co., Ltd. (Anhui, China), respectively.

### 3.2. Plant Material

Comminuted aerial parts of *E. sagittatum* (Berberidaceae) were collected from Fenghui Epimedium herb GAP Base, Zhumadian, Henan Province, People’s Republic of China, in September 2020. A voucher specimen (no. 20200960) was deposited in the Department of Pharmacy, Henan University of Chinese Medicine.

### 3.3. Extraction and Isolation

Air-dried aerial parts of *E. sagittatum* (80 kg) were extracted with 70% EtOH in three cycles, yielding an extract weighing 6.5 kg. Then, the extract was further extracted with CH_2_Cl_2_ and EtOAc. Subsequently, the CH_2_Cl_2_ fraction (2.1 kg) was subjected to silica gel column chromatography (CC, 100–200 mesh) and eluted with a petroleum ether/ethyl acetate mixture (*v*:*v*, 50:1–0:1) to afford eight fractions (Fr. A–H).

Fraction G (100.0 g) was subjected to silica gel CC (100–200 mesh) with petroleum ether/EtOAc (*v*:*v*, 50:1–0:1) to afford eighteen subfractions, Fr. G1–G18. Then, Fr. G17 (65.0 g) was separated by silica gel CC (200–300 mesh) to obtain nineteen fractions, Fr. G17.1–G17.19. Subsequently, Fr. G17.15 (38.9 g) was separated by ODS CC (MeOH:H_2_O = 40:60–100:0) to obtain twelve fractions, Fr. G17.15.1–G17.15.12. Further purification of Fr. G17.15.10 by semipreparative HPLC resulted in the isolation of compound **1** (3.3 mg, t_R_ 99.0 min, CH_3_CN–H_2_O, 65:35, 2.5 mL/min). Fr. G17.15.9 was purified by semipreparative HPLC to obtain compounds **2** (5.2 mg, t_R_ 39.2 min, CH_3_CN–H_2_O, 70:30, 2 mL/min), **3** (8.2 mg, t_R_ 55.8 min, CH_3_CN–H_2_O, 45:55, 2 mL/min), and **4** (4.0 mg, t_R_ 39.2 min, CH_3_CN–H_2_O, 80:20, 2 mL/min).

Epimesatine P (**1**): yellow amorphous powder, [α]^20^_D_ -2.3 (c 0.13, MeOH); UV (MeOH) λ_max_ (log ε): 205 (4.54), 268 (4.15), 343 (4.27) nm; IR (ν_max_): 3392, 2980, 1653, 1607, 1474, 1439, 1365, 1166, 1043, 843 cm^−1^; ^1^H and ^13^C NMR data, see Table 1; HRESIMS m/z 477.1509 [M + Na]^+^ (calcd. for C_25_H_26_O_8_Na, 477.1520).

Epimesatine Q (**2**): yellow amorphous powder, [α]^20^_D_ +5 (c 0.1, MeOH); UV (MeOH) λ_max_ (log ε): 202 (4.54), 268 (4.13), 342 (4.30) nm; IR (ν_max_): 3414, 2978, 1652, 1606, 1474, 1438, 1362, 1165, 1046, 841 cm^−1^; ^1^H and ^13^C NMR data, see Table 1; HRESIMS m/z 549.2083 [M + Na]^+^ (calcd. for C_29_H_34_O_9_Na, 549.2095).

Epimesatine R (**3**): yellow amorphous powder, [α]^20^_D_ -5.6 (c 0.2, MeOH); UV (MeOH) λ_max_ (log ε): 208 (4.53), 268 (4.13), 341 (4.29) nm; IR (ν_max_): 3432, 2979, 1653, 1612, 1474, 1439, 1362, 1165, 1063, 841 cm^−1^; ^1^H and ^13^C NMR data, see Table 1; HRESIMS m/z 565.2036 [M + Na]^+^ (calcd. for C_29_H_34_O_10_Na, 565.2044).

Epimesatine S (**4**): yellow amorphous powder, UV (MeOH) λ_max_ (log ε): 201 (4.45), 272 (4.15), 327 (4.08) nm; IR (ν_max_): 3360, 2968, 2930, 1658, 1615, 1431, 1368, 1259, 1033, 846 cm^−1^; ^1^H and ^13^C NMR data, see Table 1; HRESIMS m/z 421.2006 [M + H]^+^ (calcd. for C_26_H_29_O_5_, 421.2010).

### 3.4. ECD Calculations

The conformations of **1**–**3** were analyzed by GMMX software (version 3.0.0) using a MMFF94 force field. Geometry optimizations and predictions of the ECD spectra of the conformers were conducted through density functional theory (DFT) at the B3LYP/6-31G(d) level using Gaussian 16W software. ECD curves were simulated by SpecDis software (version 1.71) based on the Boltzmann distribution theory.

### 3.5. Preparation of the Mo_2_(OAc)_4_ Complex of Compound **2**

Firstly, Mo_2_(OAc)_4_ (1.6 mg) was dissolved in DMSO (1 mL) at room temperature. Then, this solution was added to compound **2** (0.5 mg) and thoroughly mixed. The initial ECD spectrum was recorded immediately to establish a background absorption, followed by the recording of complex-induced ECD spectra at ten-minute intervals. The absolute configuration of **2** was determined based on the Cotton effect observed in the complex-induced ECD spectra, in accordance with the helicity rule.

### 3.6. Preparation of the Rh_2_(OCOCF_3_)_4_ Complex of Compound **3**

Firstly, Rh_2_(OCOCF_3_)_4_ (1.2 mg) was dissolved in anhydrous CH_2_Cl_2_ (800 μL) at room temperature. Then, this solution was combined with compound **3** (0.6 mg) and thoroughly mixed. The first ECD spectrum was immediately recorded as a baseline, after which complex-induced ECD spectra were recorded at ten-minute intervals until reaching a stable state. The absolute configuration of **3** was determined by the Cotton effect observed in the E bond around 350 nm in complex-induced ECD spectra.

### 3.7. In Vitro Cytotoxicity Assays

#### 3.7.1. Cell Culture

Frozen MCF-7 and MCF-10A cells were purchased from the Shanghai Cell Bank of China, and they were melted in a 37 °C water bath until the state of ice and water coexisted. Then, the mixture was centrifuged immediately at 1000 rpm for 5 min. Subsequently, the supernatant was discarded and the cells were transferred to a DMEM medium containing 10% FBS (100 kU/L for both penicillin and streptomycin). Cells were cultivated in a constant temperature incubator at 37 °C containing 5% CO_2_ until the cells grew to cover 80%~90% of the dish for passage. Medium was replaced with fresh culture medium every 24 h.

#### 3.7.2. MTT Assay

MCF-7 and MCF-10A cells were cultured until the logarithmic growth phase in an incubator containing 5% CO_2_ at 37 °C. Then, the cells were inoculated into a 96-well plate (E190236X, PerkinElmer, Waltham, MA, USA) at a density of 2 × 10^4^ cells/mL and a volume of 200 μL per well. After 24 h of incubation, these wells were divided into normal control (CON) and sample (10 μM) groups to culture for another 24 h. Subsequently, 20 μL of MTT (Solarbio life sciences, Beijing, China) solution (5 mg/mL) was added to each well and cells continued to be cultured for 4 h. Then, the culture medium was aspirated carefully and 150 μL of DMSO was added to each well to dissolve the blue-violet crystals. The OD values of each well were measured by a microplate reader at 490 nm, and cell viability was calculated.

#### 3.7.3. Cellular Immunofluorescence

The assay for cellular immunofluorescence was conducted in 96-well plates, with MCF-7 cells seeded at a density of 2 × 10^4^ cells/mL. After 24 h, the cells were divided into control (CON) and various sample groups (10 μM) and cultured for an additional 24 h. The cells were then fixed with 4% paraformaldehyde for 15 min and permeabilized with 0.25% Triton X-100 for 10 min. Following this, 1% BSA was added for blocking for 30 min, after which the primary antibody Sphk1 (ab262697, Abcam) was introduced and incubated overnight at 4 °C. The cells were then washed three times with PBST, counterstained with DAPI for 4 min, washed once with PBS, and imaged using an OperettaCLS high-content imaging analysis system (Opera Phenix, PerkinElmer, Waltham, MA, USA).

#### 3.7.4. Real-Time Cellular Analysis (RTCA)

Baseline measurements were conducted using culture medium. MCF-7 cells in a logarithmic growth phase were seeded into E-plate plates at a density of 2 × 10^4^ cells/mL per well. After 24 h of incubation, these wells were divided into a CON group and sample (1 μM, 5 μM, 10 μM, 20 μM, 50 μM, and 100 μM) groups, and their growth curves were continuously measured. Upon completion, a sigmoidal dose–response (Vanable slope) algorithm was employed to determine the target time for calculating the IC_50_ values of the tested compounds. Docetaxel (Shanghai yuanye Bio-Technology Co., Ltd., Shanghai, China) served as a positive control.

#### 3.7.5. Statistical Analysis

The experimental data were expressed as mean ± standard deviation (SD) and analyzed using SPSS 26.0 software. One-way analysis of variance was used for comparison between groups. * *p* < 0.05 indicates a significant difference, while ** *p* < 0.01 indicates an extremely significant difference.

## 4. Conclusions

In summary, four previously undescribed flavonoids, epimesatines P, Q, R, and S, were isolated and identified from *E. sagittatum*. Notably, epimesatines Q and R are two flavonoids that infrequently incorporate furan rings in their structures. Cytotoxicity assays demonstrated that these compounds significantly inhibit the viability of MCF-7 human breast cancer cells while exhibiting no significant toxicity towards MCF-10A human breast epithelial cells. This suggests that these four compounds may represent the active substances of *E. sagittatum* in combating human breast cancer and demonstrate favorable safety profiles. Particularly, epimesatines Q and R exhibited greater efficacy than epimesatines P and S, indicating that the presence of furan rings in their structures may enhance their activity against MCF-7 cells, thereby providing a potential avenue for the discovery of more effective anti-breast cancer agents for chemists and pharmacologists. Particularly, epimesatine Q displayed a stronger inhibitory effect on MCF-7 cells compared to the positive control, docetaxel, reinforcing its potential as a promising therapeutic agent for breast cancer treatment. Furthermore, epimesatines P, R, and S were found to significantly inhibit the expression of Sphk1 in MCF-7 cells, suggesting that Sphk1 may serve as a target for breast cancer treatment. Overall, this study lays a foundational basis for the identification of lead compounds in the search for anti-breast cancer agents and provides a scientific rationale for the development and utilization of *E. sagittatum*.

## Figures and Tables

**Figure 1 molecules-29-04711-f001:**
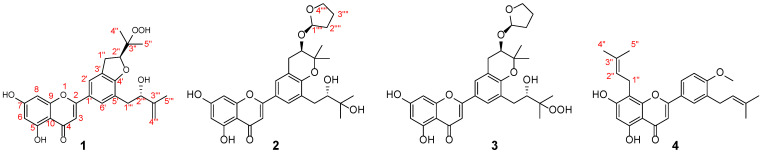
Chemical structures of epimesatines P (**1**), Q (**2**), R (**3**), and S (**4**).

**Figure 2 molecules-29-04711-f002:**
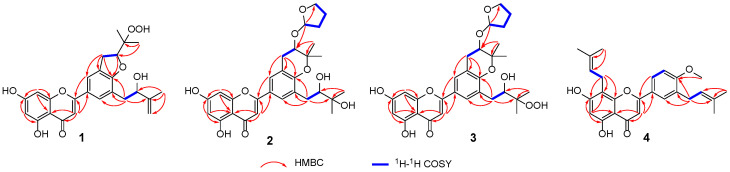
^1^H–^1^H COSY and key HMBC correlations of compounds **1**–**4**.

**Figure 3 molecules-29-04711-f003:**
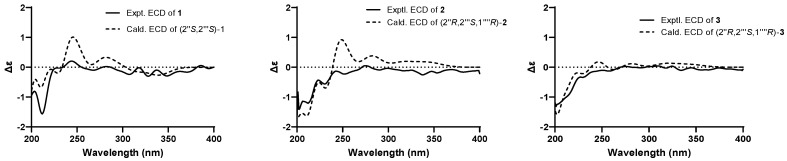
Experimental and calculated ECD spectra of compounds **1**–**3**.

**Figure 4 molecules-29-04711-f004:**
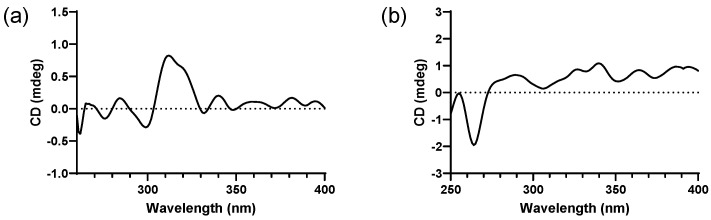
(**a**) Mo_2_(OAc)_4_−induced ECD spectrum of compound **2**; (**b**) Rh_2_(OCOCF_3_)_4_−induced ECD spectrum of compound **3**.

**Figure 5 molecules-29-04711-f005:**
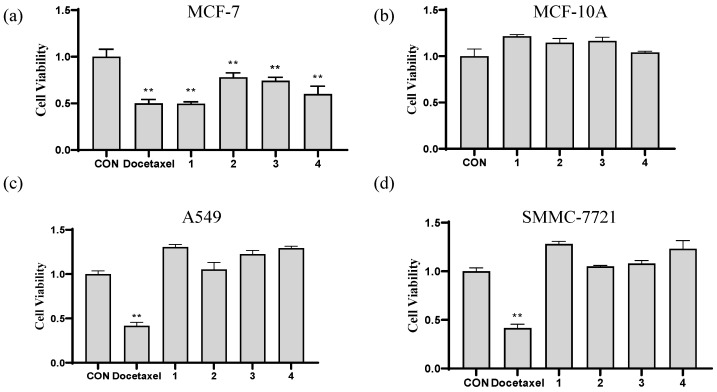
(**a**) Viability of MCF-7 human breast cancer cells after being treated by compounds **1**–**4** at a concentration of 10 μM. (**b**) Viability of MCF-10A human breast epithelial cells after being treated by compounds **1**–**4** at a concentration of 10 μM. (**c**) Viability of A549 human lung carcinoma cells after being treated by compounds **1**–**4** at a concentration of 10 μM. (**d**) Viability of SMMC-7721 human hepatoma cells after being treated by compounds **1**–**4** at a concentration of 10 μM. Docetaxel was used as a positive control. *** p* < 0.01 compared to the CON group. Each bar and vertical line represent the mean ± SD of the values from three independent experiments.

**Figure 6 molecules-29-04711-f006:**
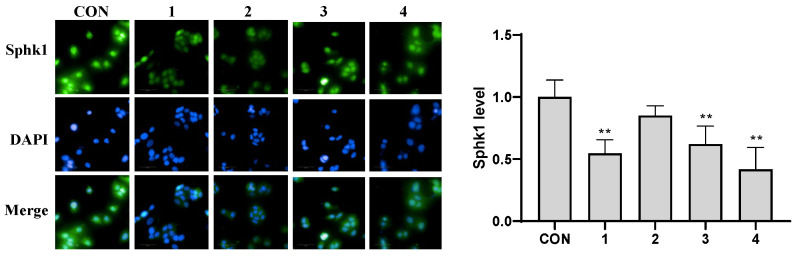
The effects of compounds **1**–**4** on the expression of Sphk1 in MCF-7 cells. ** *p* < 0.01 compared to the CON group. Each bar and vertical line represent the mean ± SD of the values from three independent experiments.

**Table 1 molecules-29-04711-t001:** ^1^H (500 MHz) and ^13^C (125 MHz) NMR data of **1**–**4** in acetone-*d*_6_ (*δ* in ppm, *J* in Hz).

No.	1		2		3		4	
	*δ* _H_	*δ* _C_	*δ* _H_	*δ* _C_	*δ* _H_	*δ* _C_	*δ* _H_	*δ* _C_
2		165.4		165.1		165.4		164.9
3	6.60, s	104.2	6.57, s	104.2	6.61, s	104.2	6.65, s	104.2
4		182.9		182.8		182.9		183.3
5		163.1		163.0		163.1		160.7
6	6.25, d (2.0)	99.7	6.24, s	99.9	6.24, s	99.6	6.34, s	99.2
7		165.1		165.7		165.0		162.1
8	6.54, d (2.0)	94.8	6.54, s	95.0	6.53, s	94.8		107.4
9		158.8		158.9		158.9		156.0
10		105.2		104.9		105.3		105.4
1′		124.1		123.1		123.0		124.3
2′	7.74 ^a^	122.1	7.64, d (2.1)	127.3	7.65, d (2.2)	127.3	7.87, d (2.3)	128.1
3′		128.9		120.8		121.6		131.6
4′		162.6		155.5		155.3		161.4
5′		122.5		130.1		130.1	7.16, d (8.7)	111.7
6′	7.75 ^a^	129.7	7.71, d (2.1)	128.5	7.73, d (2.2)	128.4	7.93, dd (8.7, 2.3)	126.9
1′‘	3.33, m	31.1	3.15, dd (16.8, 5.0)	27.8	3.12, dd (16.5, 5.3)	32.2	3.57, d (7.2)	22.4
			2.91, dd (16.8, 6.4)		2.84, dd (16.5, 7.9)			
2′‘	5.10, dd (9.8, 7.7)	87.4	3.90 ^a^	71.6	3.88 ^a^	69.3	5.32, m	123.4
3′‘		83.0		78.4		79.1		132.1
4′‘	1.36, s	21.5	1.38, s	26.4	1.43, s	26.3	1.66, s	25.9
5′‘	1.19, s	19.6	1.32, s	22.6	1.33, s	21.1	1.82, s	18.2
1′‘‘	2.97, dd (13.7, 4.9)	37.2	3.11, d (13.5)	33.9	2.94, dd (13.5, 1.8)	33.5	3.39, d (7.5)	29.1
	2.74, dd (13.7, 8.2)		2.45, dd (13.5, 10.1)		2.38, dd (13.5, 10.0)			
2′‘‘	4.42, dd (8.2, 4.9)	75.2	3.62, d (10.1)	78.3	4.03, dd (10.0, 1.8)	72.9	5.36, m	122.7
3′‘‘		149.0		72.9		86.0		133.7
4′‘‘	4.90, s; 4.75, s	110.6	1.26, s	26.1	1.38, s	22.3	1.75, s	25.9
5′‘‘	1.83, s	18.1	1.24, s	24.8	1.13, s	18.5	1.75, s	17.9
1′‘‘‘			5.38, d (4.5)	101.2	5.53, dd (6.2, 2.4)	106.4		
2′‘‘‘			1.88 ^a^	33.1	2.06 ^a^	29.7		
			1.79 ^a^		1.72, m			
3′‘‘‘			1.98 ^a^	24.1	1.91, m	24.7		
			1.85 ^a^		1.80, m			
4′‘‘‘			3.86 ^a^	67.5	3.79 ^a^	68.3		
					3.76 ^a^			
OCH_3_							3.97, s	56.2

s: singlet, d: doublet, dd: double doublet, m: multiplet. ^a^ Signals partially overlapped.

**Table 2 molecules-29-04711-t002:** The effects of compounds **1**–**4** on MCF-7 cells.

Group	1	2	3	4	Docetaxel ^#^
IC_50_ (μM)	50.3	1.27	4.59	6.05	2.13

^#^ Docetaxel was used as a positive control.

## Data Availability

Data are contained within the article and Appendix A.

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
