# Peer review of "Epimesatines P–S: Four Undescribed Flavonoids from Epimedium sagittatum Maxim. and Their Cytotoxicity Activities"

_molecules, 2024, doi:10.3390/molecules29194711_

Round 1

Reviewer 1 Report

Comments and Suggestions for Authors

Abstract: ''epimesatines P–S(1–4)'' should be better explained, more detailed, and informative (for example, all individual names because 1-4 or P-S are not informative); via should be in italics.

Introduction: all previously tested cancer cells should be mentioned (line 49); the plant should be explained in more detail (botanical characteristics, habitat, all biologically active compounds, biological activities, potential products on the market, etc.); the authors should avoid using ''we...''; the title of Figure 1 should be improved with important information such as compounds' name. The authors should better point out the novelty of their research.

Results: line 87, EDC-full name should be provided; all titles' should have the full meaning of all used abbreviations; used statistical tools and statistics should be given in the title of the figures (missing in Fig 6). Statistics are missing on the graph of Fig 5b. 

M&M: The authors should mention the pretreatment of dried plant material (potential fragmentation, comminution, etc.); producer and country should be provided for all reagents; the full Latin name should be mentioned only in the abstract, introduction, and titles, after that, the author should use E. sagittatum; 

Conclusion: the values, such as concentrations, should not be in the conclusion section; future perspectives and plans should be given.

Reviewer 2 Report

Comments and Suggestions for Authors

 In this manuscript, the authors reported four previously undescribed flavonoids from Epimedium sagittatum Maxim and their cytotoxity activities. Please consider the suggested comments to improve the quality of the current version of the manuscript:

1    In the introduction, please include the knowledge gaps existing between the current proposed study and prior studies performed in the field. Also, please include additional details on flavonoids and cytotoxity activities, to guide the reader to understand the importance of the study performed.

 2  Ln 155, please include the scientific reasoning for your observations, especially for cytotoxity activities and possible pathways. 

 3. The current version of  "Results and Discussion" is only explaining and the results instead of providing scientific reasoning. In general, it is written a little illegibly and needs thorough revision. Kindly incorporate more references in the results and discussion session

 4 Ln 172, lack of the detailed information and relevant references.

 5 Ln 180, lack of the detailed information about methods.

 6 The conclusion section was not sufficient to effectively summarize the content of the entire manuscript. This section needs to be more specific and summarized.

Comments on the Quality of English Language

English language is fine.

Reviewer 3 Report

Comments and Suggestions for Authors

IN THIS current PAPER new rare including furan rings flavonoids, named epimesatines were isolated from the aerial parts of Epimedium sagittatum Maxim. Their structures and absolute configurations were obtained via spectroscopic analyses, quantum chemical electronic circular dichroism (ECD) calculations, Mo2(OAc)4-induced ECD, and Rh2(OCOCF3)4-induced ECD experiments. The cytotoxicity assay demonstrated that epimesatines P–S exhibited significant inhibitory effects on the viability of MCF-7 human breast cancer cells, with IC50 values ranging from 1.27 to 50.3 μM. However none of the tested compounds exhibited obvious toxicity towards MCF-10A human breast epithelial cells. Furthermore, compounds 1, 3, and 4 were found to significantly inhibit the expression of sphingosine kinase 1 (Sphk1) in MCF-7 cells

Comments good paper, goods results, spectroscopic evidence of new compounds, with 2 d and 1 d NMR spectra as supp material, supporting the findings, tables and diagrams results are good, but I would add more cells to test. Please check carefully English language. It should be good if possible to add some other interesting bioactivity. Please add pictures of plant material

Comments on the Quality of English Language

good

Round 2

Reviewer 2 Report

Comments and Suggestions for Authors

Accept in present form.